# Immunogenicity Assessment of the SARS-CoV-2 Protein Subunit Recombinant Vaccine (CoV2-IB 0322) in a Substudy of a Phase 3 Trial in Indonesia

**DOI:** 10.3390/vaccines12040371

**Published:** 2024-04-01

**Authors:** Sharifah Shakinah, Muhammad Hafiz Aini, Rini Sekartini, Bernie Endyarni Medise, Hartono Gunardi, Irene Yuniar, Wahyuni Indawati, Sukamto Koesnoe, Kuntjoro Harimurti, Suzy Maria, Angga Wirahmadi, Rini Mulia Sari, Lilis Setyaningsih, Fikrianti Surachman

**Affiliations:** 1Department of Internal Medicine, Faculty of Medicine, Universitas Indonesia, Dr. Cipto Mangunkusumo General National Hospital, Jalan Diponegoro No 71, Jakarta 10340, Indonesia; sharifah.shakinah@gmail.com (S.S.); suzyduri@gmail.com (S.M.); 2Department of Internal Medicine, Universitas Indonesia Hospital, Jl. Prof. DR. Bahder Djohan, Depok 16424, Indonesia; 3Department of Child Health, Faculty of Medicine, Universitas Indonesia, Dr. Cipto Mangunkusumo General National Hospital, Jalan Diponegoro No 71, Jakarta 10340, Indonesia; rsekartini@yahoo.com (R.S.); soedjat3@yahoo.com (S.); wahyuni_indawati@yahoo.com (W.I.);; 4PT Bio Farma, Jalan Pasteur No. 28, Bandung 40161, Indonesialilis.setyaningsih@biofarma.co.id (L.S.); fikrianti.surachman@biofarma.co.id (F.S.)

**Keywords:** CoV2-IB 0322 vaccine, T-cell response, B-cell response

## Abstract

Background: COVID-19 is one of the most devastating pandemics of the 21st century. Vaccination is one of the most effective prevention methods in combating COVID-19, and one type of vaccine being developed was the protein subunit recombinant vaccine. We evaluated the efficacy of the CoV2-IB 0322 vaccine in Depok, Indonesia. Methods: This study aimed to assess the humoral and cellular immune response of the CoV2-IB 0322 vaccine compared to an active control vaccine (COVOVAX™ Vaccine). A total of 120 subjects were enrolled and randomized into two groups, with 60 subjects in each group. Participants received either two doses of the CoV2-IB 0322 vaccine or two doses of the control vaccine with a 28-day interval between doses. Safety assessments were conducted through onsite monitoring and participant-reported adverse events. Immunogenicity was evaluated by measuring IgG anti-RBD SARS-CoV-2 and IgG-neutralizing antibodies. Cellular immunity was assessed by specific T-cell responses. Whole blood samples were collected at baseline, 14 days, 6 months, and 12 months after the second dose for cellular immunity evaluation. Results: Both vaccines showed high seropositive rates, with neutralizing antibody and IgG titers peaking 14 days after the second dose and declining by 12 months. The seroconversion rate of anti-S IgG was 100% in both groups, but the rate of neutralizing antibody seroconversion was lower in the CoV2-IB 0322 vaccine group at 14 days after the second dose (*p* = 0.004). The CoV2-IB 0322 vaccine showed higher IgG GMT levels 6 and 12 months after the second dose (*p* < 0.001 and *p* = 0.01). T-cell responses, evaluated by IFN-γ, IL-2, and IL-4 production by CD4+ and CD8+ T-cells, showed similar results without significant differences between both groups, except for %IL-2/CD4+ cells 6 months after the second dose (*p* = 0.038). Conclusion: Both vaccines showed comparable B- and T-cell immunological response that diminish over time.

## 1. Introduction

Coronavirus disease 19 (COVID-19) is an infectious disease caused by the SARS-CoV-2 virus. Since the publication of the first case in Hubei, China, COVID-19 has reached 772 million confirmed cases and caused 6.9 million deaths [1]. COVID-19 is one of the most devastating pandemics of the 21st century, profoundly impacting global health, economies, and societies worldwide [2]. Although the emergency phase of COVID-19 is over, several preventive measures still need to be taken and maintained to protect high-risk populations. Vaccination is one of the most effective methods in preventing COVID-19 infection [3]. The World Health Organization (WHO) has approved ten vaccines for emergency and full use. Different types of COVID-19 vaccines have been studied and used in daily practice, such as messenger RNA (mRNA) vaccines, vector vaccines, inactive vaccines, and protein subunit vaccines. Each COVID-19 vaccine induces the immune system to create immune responses to combat the virus with a different approach [4,5].

A protein subunit vaccine uses microorganisms’ fragments as the pathogen antigenic components to induce effective immune responses [6]. SARS-CoV-2 possesses both structural and non-structural proteins. The primary structural proteins include the spike (S), membrane (M), and envelope (E) proteins, situated within the phospholipid bilayer, and the nucleocapsid (N) protein. The S protein plays a pivotal role in mediating viral attachment and entry into the host cell by interacting with the angiotensin-converting enzyme-2 (ACE2 receptor). Structurally, the S protein comprises two subunits, S1 and S2, with the S1 subunit responsible for receptor recognition, while the S2 subunit plays an important role in membrane fusion [7].

The CoV2-IB 0322 vaccine, a protein recombinant subunit vaccine engineered by PT Bio Farma. utilizes the RBD from the S protein of the SARS-CoV-2 strain as its antigen. This vaccine formulation includes adjuvants such as aluminum hydroxide and CpG1018. Adjuvants play an important role in eliciting specific immune responses. Phase 1 and 2 studies of the CoV2-IB 0322 vaccine were done to determine the optimal dosage. The final formulation has shown the lowest side effect with comparable promising efficacy. The optimum formulation showed a good humoral immune response, as reflected in the anti-RBD IgG titer and neutralizing antibody titers, with minimal side effects and without any severe adverse effects reported. The RBD protein clones used in this vaccine were developed by Texas Children’s Hospital Center for Vaccine Development (TCH-CVD) at Baylor College of Medicine (BCM), USA). These clones were based on the amino acid sequence of the wild-type SARS-CoV-2 RBD amino acid, representing residues 331–549 of the spike (S) protein (GenBank: QHD43416.1) of the Wuhan-Hu-1 isolate (GenBank: MN908947.3) [8].

The ongoing COVID-19 pandemic, characterized by the emergence of various viral variants, highlights the need for versatile and quickly adaptable vaccines. Recombinant protein vaccines offer a promising alternative due to their ability to be rapidly developed and produced, potentially outpacing the evolution of the virus. The flexibility and speed of recombinant vaccine technology could be crucial in addressing the current challenges posed by COVID-19 and future pandemics, enabling quicker adaptation to new viral threats, and facilitating widespread immunization efforts [9,10].

This study is a part of the phase III randomized control trial of SARS-CoV-2 Protein Subunit Recombinant Vaccine (Bio Farma, Bandung, Indonesia) Adjuvanted with Alum+CpG 1018 compared to the Registered COVID-19 Vaccine (COVOVAX™ Vaccine—Protein Subunit Vaccine). It aimed to evaluate the immunogenicity of the SARS-CoV-2 Protein Subunit Recombinant Vaccine (CoV2-IB 0322) and to assess humoral and cellular immunity of the vaccine at 14 days, 6 months, and 12 months after the primary series of vaccination. Evaluating humoral and cellular immune response to SARS-CoV-2 after vaccination is important to determine whether protective immunity is sustained over the long-term following vaccination [11].

## 2. Materials and Methods

### 2.1. Study Design and Participants

This study is a part of the phase 3 observer-blind, active-controlled, prospective intervention study of comparing the SARS-CoV-2 Protein Subunit Recombinant Vaccine (CoV2-IB 0322) adjuvanted with Alum+CpG 1018, produced by PT Bio Farma, with the registered COVID-19 Vaccine (COVOVAX™ Vaccine—Protein Subunit Vaccine) in healthy adults. The exploratory study subset was conducted in Puskesmas Duren Seribu, Puskesmas Bojongsari, and Puskesmas Pasir Putih Depok, Indonesia from July 2022 to July 2023. This study has been registered on ClinicalTrials.gov under the trial registration number NCT05433285.

All participants provided written informed consent before enrollment into the study. The inclusion criteria were healthy subjects aged 18 years and above who committed to complying with study instructions and study schedules. The exclusion criteria were participants who had enrolled in another trial, had prior COVID-19 vaccination, had prior COVID-19 infection (mild-to-moderate disease within 1 month or severe disease within 3 months), had fever prior to inclusion, were pregnant or planning to become pregnant during the study period, had a history of blood disorders that could cause contraindication of intramuscular injection, had serious chronic diseases that might disturb assessment of the trial objective, had a history of confirmed or suspected immunosuppressive or immunodeficient states, had received treatment likely to alter the immune response, had a history of uncontrolled epilepsy or other progressive neurological disorders, had received any other vaccination within 1 month, or were planning to leave the study area before the trial was completed.

A total of 120 subjects were divided into two groups, the vaccine group and the active control (COVOVAX™ Vaccine) group, with 60 subjects in each group. 60 subjects per arm received the SARS-CoV-2 subunit protein recombinant vaccine, and 60 subjects received the active control vaccine. Randomization was done using a computer-generated randomization list with a 1:1 ratio. While gender was not a stratifying factor in the randomization algorithm, this approach was chosen to ensure fairness and prevent bias in allocation to treatment groups. Immunogenicity and cellular immunity were the primary focus of analysis for all participants. Safety protocols were monitored continuously during the study.

### 2.2. Procedures

Two doses of the SARS-CoV-2 vaccine or two doses of the control (two doses of the COVOVAX™ Vaccine—SARS-CoV-2 rS Protein (COVID-19) recombinant spike protein Nanoparticle Vaccine) were administered with a 28 day-interval between each dose. For each dose administration, participants were injected with a 0.5 mL dose of the study vaccine or control vaccine intramuscularly. Onsite monitoring was done at least 30 min after dose administration.

Participants were provided with a diary card to record any local or systemic adverse events occurring 28 days following each dose of the vaccine. Blood samples were taken to evaluate IgG anti-RBD SARS-CoV-2 and IgG-neutralizing antibodies of SARS-CoV-2. The quantification of IgG antibodies specific to the SARS-CoV-2 spike protein was performed using an enzyme-linked immunosorbent assay (ELISA). Neutralizing antibody titers were determined using a pseudovirus neutralization assay. This involved mixing serial dilutions of participant serum with lentiviral particles bearing the SARS-CoV-2 spike protein, then incubating this mixture with Vero E6 cells. After 72 h, the extent of infection was assessed via luciferase activity, with the neutralizing antibody titer identified as the serum dilution yielding a 50% reduction in this activity.

Cellular immunity was evaluated by venous blood sampling that was drawn on Day 0, Day 42, Day 208, and Day 388. Monthly follow-ups by phone were done to monitor subjects. Specific T-cell responses (CD4+ and CD8+-producing IFN-γ, IL-2, and IL-4) at 14 days, 6 months, and 12 months after the two-dose primary series were evaluated, along with the Geometric Mean Titer (GMT) and seropositive rate of SARS-CoV-2 (RBD)-binding IgG antibody and seropositive rate of the neutralizing antibody. We also evaluated the seroconversion rate of the SARS-CoV-2 (RBD)-binding IgG antibody and the seroconversion rate of the neutralizing antibody at 14 days after the two-dose primary series.

For cellular immunity evaluation, whole blood samples were taken at baseline (V1), 14 days (V2a), 6 months (V3b), and 12 months (V4) after the second dose and further processed into Peripheral Blood Mononuclear Cells (PBMCs). Cells were then stained with fluorescently labeled antibodies against CD4, CD8, and additional markers for T-cell activation and function. After staining, cells were fixed, permeabilized, and stained for intracellular cytokines (IFN-gamma, IL-2, IL-4) using specific antibodies. The cytokines that were induced by peptides representing the vaccine-encoded RBD were measured as a percentage of RBD-specific CD8+ and CD4+ T-cells by the evaluation of intracellular cytokines (ICS) and cell-mediated immune response (CBA). Full laboratory methodologies are available upon request.

### 2.3. Vaccine and Control

The randomization was assigned by an unblinded team, which held the generated randomization list. Subjects were divided into two groups, the CoV2-IB 0322 vaccine group and the COVOVAX™ Vaccine group as an active control. To maintain observer-blindness, several measures were adopted throughout the study. Vaccines for both the study and control groups were prepared in identical syringes by personnel not involved in the administration or assessment of study outcomes. Healthcare providers administering the vaccines and participants were blinded to group assignments. Additionally, a separate team, unaware of the treatment allocations, was responsible for assessing study outcomes. All data analysis was conducted on anonymized datasets to further ensure that the observer-blindness was preserved until the study’s conclusion.

The vaccine studied was the CoV2-IB 0322 vaccine, a protein recombinant subunit vaccine developed by PT Bio Farma. The final formulation consisted of 25 μg of SARS-CoV-2 RBD subunit recombinant protein, 750 μg of aluminum as an adjuvant, 750 μg of CpG 1018 as an adjuvant, 2.226 mg of NaCl, and 0.923 mg of tris(hydroxymethyl) aminomethane administered by intramuscular injection, chosen based on phase 1/2 studies.

The COVOVAX™ Vaccine, a SARS-CoV-2 rS Protein (COVID-19) recombinant spike protein Nanoparticle Vaccine, served as the active control. Each 0.5 mL dose comprised SARS-CoV-2 recombinant spike protein (5 μg per dose) with Matrix-M adjuvant (50 μg per dose). The inactive ingredients are sodium chloride, disodium hydrogen phosphate dibasic heptahydrate, sodium dihydrogen phosphate monohydrate, and Polysorbate 80. This vaccine was developed by Serum Institute of India Pvt. Ltd., Pune, India. The active control was considered more ethical than a placebo to protect controlled subjects from getting infected, as the study was done during a time when vaccination for COVID-19 was mandatory.

### 2.4. Statistical Analysis

Statistical analysis was done using Chi-square for categorical outcomes, such as seropositive and seroconversion between groups, and the ANOVA test for numeric outcomes, such as GMT of IgG anti-RBD SARS-CoV-2 and cytokine levels. Comparison of the geometric mean of antibody titer was run after the data were log-transformed. Statistical analysis was done using SPSS 20.0.

The Full Analysis Set (FAS) included all participants who were randomized, adhered to the intention-to-treat (ITT) principle, and possessed valid immunogenicity data prior to vaccination. The Per-Protocol Set (PPS) consisted of those participants who were randomized, fulfilled the inclusion criteria, avoided the exclusion criteria, received the booster vaccination in full, and had valid immunogenicity data before and after vaccination. Immunogenicity analyses utilized the PPS.

## 3. Results

From 29 June to 27 July 2022, 120 subjects were screened in three primary health centers in Depok. The Jakarta Centre was responsible for an exploratory subset of patients in the CoV2-IB 0322 vaccine trial. 120 subjects were recruited and randomized into two groups as shown in Figure 1. In the CoV2-IB 0322 vaccine group, 60 subjects received the first dose, 55 completed 12 months of follow-up; five patients were withdrawn (three dropped out; two lost to follow-up). In the COVOVAX™ Vaccine group, 60 subjects received the first dose, and 54 subjects completed 12 months of follow-up; six were withdrawn (three dropped out; two lost to follow-up; one discontinued by the investigator).

Table 1 shows the characteristics of research subject. Hypertension was the most prevalent comorbidity found in our study (16 subjects). Other comorbidities were asthma (two subjects), diabetes mellitus (two subjects), and rheumatoid arthritis (one subject). No significant differences in comorbidities were observed between both groups.

Humoral and cellular immunity responses were evaluated in 120 participants. For cellular immunity evaluation, whole blood samples were taken at baseline (V1), 14 days (V2a), 6 months (V3b), and 12 months (V4) after the second dose. Blood samples were processed into peripheral blood mononuclear cells (PBMCs). The cytokine induced by peptides representing the vaccine-encoded RBD were measured as a percentage of RBD-specific CD8+ and CD4+ T-cells, by evaluating intracellular cytokines (ICS) and cell-mediated immune response (CBA).

### 3.1. Neutralizing Antibody

Neutralizing antibody titers were evaluated against the Delta strain at baseline, 14 days, 6 months, and 12 months after the second dose (Table 2). In the vaccine group, the Geometric Mean Titer (GMT) in International Units per milliliter (IU/mL) at baseline, 14 days, 6 months, and 12 months after the second dose were 125.37, 2109.62, 1282.33, and 736.51, respectively. The GMT (IU/mL) in the control group were 145.20, 3262.58, 970.50, and 1096.45. No significant differences were observed between both groups. COVOVAX™ Vaccine exhibited a higher seropositive rate, but there were no significant differences in the GMTs of neutralizing antibodies between COVOVAX™ Vaccine and the CoV2-IB 0322 vaccine at 6 and 12 months after the second dose.

### 3.2. IgG Antibody

The IgG antibody as a secondary endpoint is as follows (Table 3). In the vaccine group, the GMT (Binding Antibody Units per milliliter [BAU/mL]) at baseline, 14 days, 6 months, and 12 months after the second dose were 35.45, 2320.93, 1171.88, and 513.03, respectively. In the control group, the GMT (BAU/mL) were 39.78, 2619.52, 448.22, and 380.35. Before vaccination, there were no significant differences in the seropositive rates between COVOVAX™ Vaccine and the CoV2-IB 0322 vaccine (*p* = 0.528). Both vaccines showed high seropositive rates, indicating the presence of pre-existing immunity in some individuals. At subsequent time points (14 days after the second dose, 6 months after the second dose, and 12 months after the second dose), the seropositive rates remained high for both vaccines, with the CoV2-IB 0322 vaccine showing slightly higher GMT of IgG levels at 6 and 12 months after the second dose.

### 3.3. Intracellular Cytokines (ICS)

The analysis began by gating the total CD4/CD8 T-cell populations based on CD4/CD8 expression, alongside a side scatter profile characteristic of lymphocytes. Within this context, our observations revealed an increase in IL-2 and IL-4-secreting CD4+ T-cells, alongside a decrease in IFN-gamma-secreting CD4+ T-cells in both the CoV2-IB 0322 vaccine and COVOVAX™ Vaccine groups 14 days after administering the second dose, compared to baseline levels. When comparing the two vaccine groups, we noted no significant differences in the cytokines evaluated, as outlined in Table 4 and illustrated in Figure 2 and Figure 3. It is noteworthy that the IL-2 secreting CD4+ and CD8+ T-cell levels were sustained up to 12 months post-vaccination, whereas the numbers of other T-cell types experienced a decline.

## 4. Discussion

The global spread and ongoing challenges of COVID-19 infection underscore the urgent need for effective vaccination strategies [1]. To safeguard high-risk groups, several preventive actions still need to be implemented and maintained even after the COVID-19 emergency period has ended. Vaccination is one of the most effective methods in preventing COVID-19 infection. Vaccines lead to less severe infections, therefore causing reduced COVID-19 transmission. The World Health Organization (WHO) has approved ten vaccines for emergency and full use [4]. Different types of COVID-19 vaccines have been studied and used in daily practice, such as messenger RNA (mRNA) vaccines, vector vaccines, inactive vaccines, and protein subunit vaccines. All reported vaccine candidates have shown promising efficacy with minimal side effects [12].

This is the first study evaluating immunologic response in patients receiving the SARS-CoV-2 protein recombinant subunit vaccine, the CoV2-IB 0322 vaccine. Immunologic response, including both B-cell associated immune responses and T-cell associated immune responses, were evaluated with several markers such as levels of neutralizing antibodies, IgG antibodies, and intracellular cytokines (IFN-gamma, IL-2, IL4 produced by CD4+ and CD8+) [13,14,15].

Neutralizing antibodies are surrogate markers that correlate with protection against SARS-CoV-2 infection. Neutralizing antibodies play a pivotal role in virus clearance and protection against SARS-CoV-2 infection and are a good biomarker of host defense through humoral immunity. Studies have shown that the absence of neutralizing antibodies correlates with mortality and delayed viral control. Severely ill patients show higher levels of neutralizing antibody titers compared to mild cases [16,17,18].

Neutralizing antibody and IgG antibody titers were evaluated against the Delta strain, showing no significant difference in baseline between the CoV2-IB 0322 vaccine and Covovax groups.

Both COVOVAX™ Vaccine and the CoV2-IB 0322 vaccine showed high seropositive rates before vaccination, indicating the presence of pre-existing immunity in a significant portion of the study population. Both neutralizing antibodies and IgG titers peaked 14 days after the second dose and declined by 12 months to approximately 30–35% of the peak level in both groups. The seroconversion rate of anti-S IgG was 100% 14 days after the second dose in both groups, but the seroconversion rate of the neutralizing antibody was lower in the CoV2-IB 0322 vaccine group than in the COVOVAX™ Vaccine group. Timing of the waning antibody levels may predict a loss of protection to the variants tested [19,20]. Several factors contribute to the dynamics of the humoral immune response. These factors, such as age, gender, comorbidities, timing of vaccination, and previous infection history, affect not only the antibody level titer but also correlate with the severity of COVID-19 [21,22,23].

The observed discrepancy between neutralizing antibody levels and IgG antibody levels in recipients of the CoV2-IB 0322 vaccine highlights the complexity of the immune response to different SARS-CoV-2 vaccines. This might indicate that the body has mounted a broad immune response to the pathogen but has not produced a strong neutralizing response. For instance, recombinant protein-based vaccines have been shown to induce strong antibody responses, including IgG, against specific antigens, which might not directly translate into neutralizing activity against the virus. This could be due to the specific conformational epitopes targeted by the elicited antibodies, which may not effectively inhibit viral entry into host cells despite high IgG levels. Moreover, the quality and functional capabilities of antibodies, such as their ability to mediate virus neutralization, depend on factors beyond mere quantity, including affinity maturation and the prevalence of certain IgG subclasses [24].

T-cell-mediated immune responses are essential in host defense, particularly in containing infection by the destruction of infected cells and destruction of intracellular pathogen, therefore reducing viral load [25]. Our study tries to evaluate T-cell response to vaccination by both CD4+ and CD8+ T-cells. CD4+T-cells play a central role in coordinating and regulating immune responses. They help activate other immune cells, including B-cells and CD8+ cytotoxic T-cells, and assist in the development of memory responses [26,27]. Meanwhile, CD8+ cytotoxic T-cells are primarily responsible for directly killing virus-infected or abnormal cells. They play a crucial role in cell-mediated immunity and the elimination of intracellular pathogens, as seen in COVID-19 pathogenesis [27,28]. By evaluating both CD4+ and CD8+ T-cell responses, we try to assess the magnitude and breadth of vaccine-induced immune response in both the CoV2-IB 0322 vaccine and COVOVAX™ Vaccine groups.

SARS-CoV-2 specific T-cell responses were measured by IFN-γ, IL-2, and IL-4, produced by CD4+ and CD8+ T-cells. IFN-γ is a key cytokine involved in the antiviral immune response by activating T-cells and inducing macrophages, while IL-2 is critical for T-cell proliferation and activation. IFN-γ produced by CD4 T-cells is involved in activating macrophages, enhancing the activity of other immune cells, and promoting the development of an effective immune response. IFN-γ produced by CD8 T-cells contributes to the ability to directly eliminate infected cells [29,30]. CD4+ cells are the primary source of IL-2 production, as well as CD8+ cells to a lesser extent. By evaluating both cytokine production capacities, we explored the vaccines’ efficacy in activating T-cells, promoting a Th1-biased response, and generating memory cells contributing to long-term protection against SARS-CoV-2 infection [31,32,33].

Prior to vaccination, the COVOVAX™ Vaccine group showed higher production of IFN-γ and IL-2 by CD4 cells compared to the CoV2-IB 0322 vaccine. This suggests the presence of pre-existing immunity or primed immune cells ready to respond to subsequent antigenic challenge that might potentially influence the speed and magnitude of the immune response following vaccination [34]. However, this difference diminished over time, with no significant disparities observed at later time points, indicating comparable T-cell activation by both vaccines in the long term.

The main functions of IL-4 are to stimulate T helper 2 (Th2) cell differentiation and proliferation and the production of IgE [35]. There were no significant differences in IL-4 production between the vaccines at any time, suggesting similar Th2 response profiles. Both vaccines induced comparable responses in CD8+ T-cells from % IFN-γ/CD8+ and % IL-2/CD8+, as evidenced by the lack of significant differences in IFN-γ and IL-2 production.

Overall, while there were some initial differences in cytokine production between COVOVAX™ Vaccine and the CoV2-IB 0322 vaccine, particularly in CD4 cell responses, these distinctions diminished over time, suggesting that both vaccines may offer similar long-term coordinated humoral and cellular immune responses. The differences observed could be important for understanding the initial immune response dynamics, but they may not necessarily translate into significant vaccine efficacy or effectiveness variations in the long run.

This is the first study to evaluate both B- and T-cell immunologic responses in the CoV2-IB 0322 vaccine. Understanding the dynamics of immune responses induced by different COVID-19 vaccines is crucial for optimizing vaccination strategies and enhancing population-wide immunity [11]. Our study was subject to certain limitations, such as the evaluation of immunologic responses done solely through blood tests without microbiological confirmation of COVID-19 infection in asymptomatic subjects. However, the findings of our study provide critical insights into the development and optimization of COVID-19 vaccines. Given the global urgency to address the pandemic and its evolving nature, our research underscores the importance of exploring diverse vaccine platforms to ensure broad and durable protection against emerging variants. Further research is warranted to evaluate the correlation between immune response profiles and clinical outcomes, such as vaccine efficacy against emerging variants and the durability of protection over time.

## 5. Conclusions

Our study evaluated humoral and cellular immunity responses in subjects who received the CoV2-IB 0322 vaccine. The CoV-2 IB 0322 vaccine showed a long-term, coordinated humoral immune response, as shown by high seropositive rates of neutralizing antibodies and high levels of IgG peaking at 14 days after the second dose of vaccine, but declining by 12 months. Cellular immune response evaluation by IFN-γ, IL-2, and IL-4 production by CD4+ and CD8+ T-cells showed a response comparable to that of active control vaccines.

## Figures and Tables

**Figure 1 vaccines-12-00371-f001:**
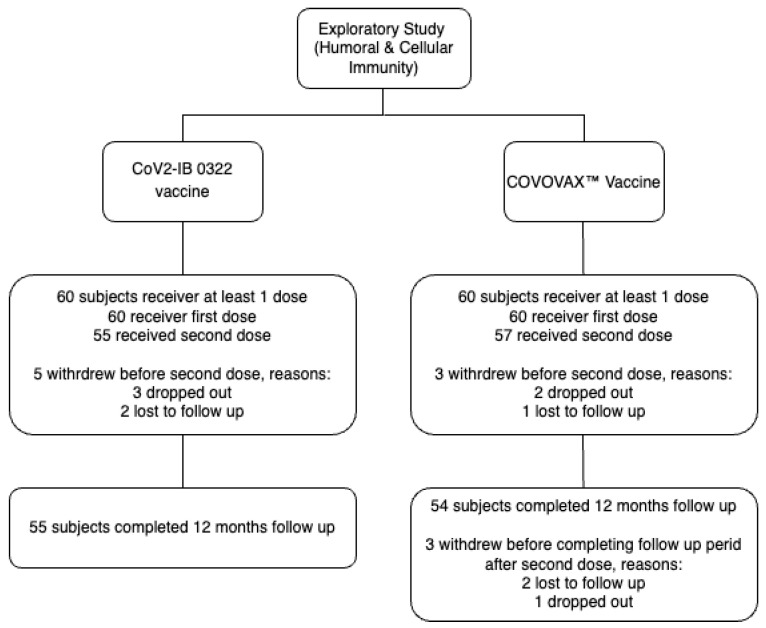
Flow of subjects included in the study.

**Figure 2 vaccines-12-00371-f002:**
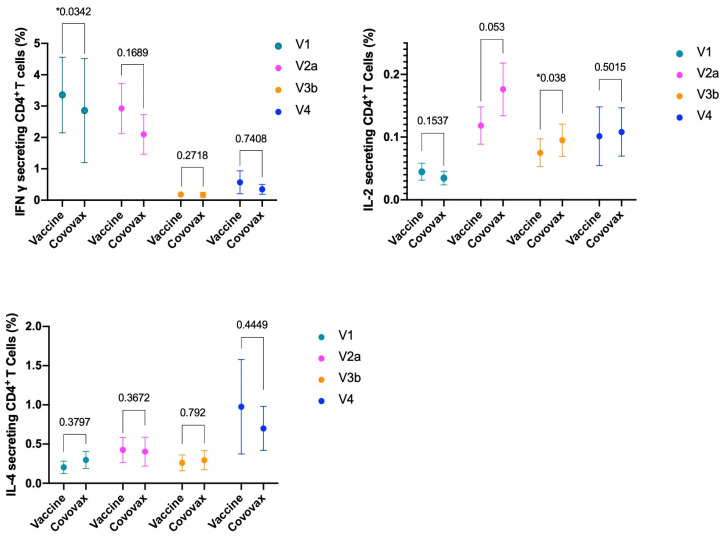
RBD-specific CD4+ T-cells evaluated by ICS (V1 = baseline; V2a = 14 days, V3b = 6 months, V4 = 12 months after the second dose). * shows statistically significant result.

**Figure 3 vaccines-12-00371-f003:**
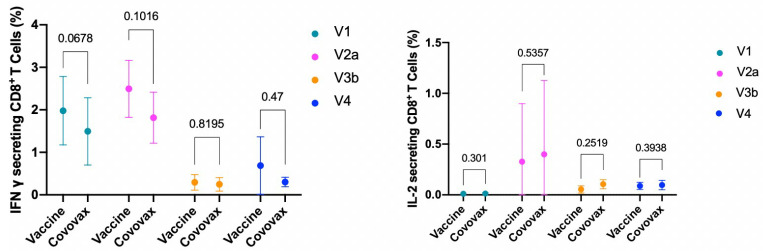
RBD-specific CD8+ T-cells evaluated by ICS (V1 = baseline; V2a = 14 days, V3b = 6 months, V4 = 12 months after the second dose).

**Table 1 vaccines-12-00371-t001:** Baseline characteristics.

	CoV2-IB 0322 Vaccine (n = 60)	COVOVAX™ Vaccine (n = 60)	*p*
Mean age/years (SD)	38.47 (14.49)	37.52 (13.09)	0.785
Sex (%) Men Women	41 (68.3) 19 (31.7)	29 (48.3) 31 (51.7)	0.041
BMI median (min–max)	23.35 (14.14–38.95)	21.7 (15.32–40.32)	0.274
Comorbidities (%) Yes No	9 (15) 51 (85)	12 (20) 48 (80)	0.471
Occupation Private employee Entrepreneur Labor Healthcare workers Other Unemployed	4 (6.7) 5 (8.3) 15 (25) 1 (1.7) 8 (13.3) 27 (45)	5 (8.3) 8 (13.3) 12 (20) 0 (0) 4 (6.7) 31 (51.7)	0.587

*p*-values of categorical outcomes are calculated with Chi-square test; *p*-values of numerical outcomes are calculated with Independent T-test and Mann–Whitney U test.

**Table 2 vaccines-12-00371-t002:** Neutralizing antibody in vaccine and control groups.

Time Point	Parameter	CoV2-IB 0322 Vaccine (n = 55)	COVOVAX™ Vaccine (n = 54)	*p*
Before Vaccination (V1)	Seropositive rate (%)	32 (58.18)	33 (61.11)	0.755
	GMT (Dilution), 95% CI	11.05 (7.02–17.39)	12.85 (7.57–21.83)	0.785
	GMT (IU/mL), 95% CI	125.37 (79.89–196.75)	145.20 (85.96–245.27)	0.785
14 days after 2nd dose (V2a)	Seropositive rate (%)	47 (85.45)	54 (100)	0.004
	GMT (Dilution), 95% CI	187.48 (104.57–336.15)	291.84 (217.94–390.82)	0.014
	GMT (IU/mL), 95% CI	2109.62 (1179.12–3774.41)	3264.58 (2437.69–4371.96)	0.014
6 months after 2nd dose (V3b)	Seropositive rate (%)	50 (90.91)	54 (100)	0.023
	GMT (Dilution), 95% CI	114.09 (69.29–187.84)	86.48 (59.15–126.44)	0.101
	GMT (IU/mL), 95% CI	1282.33 (780.41–2107.06)	970.50 (662.85–1420.93)	0.101
12 months after 2nd dose (V4)	Seropositive rate (%)	47 (85.45)	52 (96.3)	0.050
	GMT (Dilution), 95% CI	65.39 (38.31–111.59)	96.74 (62.98–148.61)	0.486
	GMT (IU/mL), 95% CI	736.51 (432.99–1252.81)	1096.45 (714.99–1681.44)	0.486

*p*-values of seropositive and seroconversion are calculated with Chi-square test; *p*-values of GMT are calculated with Mann–Whitney U Test. Note: V1 = before injection; V2a = 14 days after second injection; V3b = 6 months after second injection; V4 = 12 months after second injection; seropositive = titer ≥ 4 dilution or ≥46.03 IU/mL; seroconversion = four-fold increase compared to baseline if seropositive or a change from seronegative to seropositive.

**Table 3 vaccines-12-00371-t003:** IgG antibody in exploratory study subjects.

Time Point	Parameter	CoV2-IB 0322 Vaccine (n = 55)	COVOVAX™ Vaccine (n = 54)	*p*
Before Vaccination (V1)	Seropositive rate (%)	41 (74.55)	43 (79.63)	0.528
	IgG (AU/mL), 95% CI	249.67 (148.53–419.68)	280.18 (168.87–464.85)	0.841
	IgG (BAU/mL), 95% CI	35.45 (21.09–59.59)	39.78 (23.98–66.00)	0.841
14 days after 2nd dose (V2a)	Seropositive rate (%)	55 (100)	54 (100)	N/A ^(a)^
	IgG (AU/mL), 95% CI	16,344.46 (10,536.68–25,353.48)	18,447.30 (15,460.51–22,011.10)	0.014
	IgG (BAU/mL), 95% CI	2320.93 (1496.22–3600.20)	2619.52 (2195.39–3125.58)	0.014
6 months after 2nd dose (V3b)	Seropositive rate (%)	54 (98.18)	54 (100)	0.023
	IgG (AU/mL), 95% CI	8252.78 (5269.71–12,924.49)	3156.47 (2378.51–4188.88)	<0.001
	IgG (BAU/mL), 95% CI	1171.88 (748.28–1835.28)	448.22 (337.75–594.82)	<0.001
12 months after 2nd dose (V4)	Seropositive rate (%)	51 (92.73)	54 (100)	0.043
	IgG (AU/mL), 95% CI	3613.01 (2229.86–5854.11)	2678.50 (1948.96–3681.12)	0.015
	IgG (BAU/mL), 95% CI	513.03 (316.62–831.28)	380.35 (276.75–522.72)	0.015

*p*-values of seropositive and seroconversion are calculated with Chi-square test; *p*-values of GMT are calculated with Mann–Whitney U Test. ^(a)^ No statistics are computed because Seropositive 14-days (V2a) or Seroconversion conv V2a is a constant. Note: V1 = before injection; V2a = 14 days after second injection; V3b = 6 months after 2nd dose; V4 = 12 months after 2nd dose. Seropositive = titer ≥ 50 AU or ≥7.1 BAU; seroconversion = four-fold increase in anti-RBD antibody IgG titer compared to baseline if seropositive or a change from seronegative to seropositive.

**Table 4 vaccines-12-00371-t004:** Evaluation of Intracellular Cytokines (ICS).

Parameter	Time Point	CoV2-IB 0322 Vaccine	COVOVAX™ Vaccine	*p*
% IFNγ/CD4	Before Vaccination (V1)			
	n	52	52	
	Mean, 95% CI	3.33 (2.14–4.51)	2.86 (1.20–4.52)	0.034
	14 days after 2nd dose (V2a)			
	n	52	52	
	Mean, 95% CI	2.87 (2.07–3.66)	2.10 (1.47–2.73)	0.169
	6 months after 2nd dose (V3b)			
	n	54	53	
	Mean, 95% CI	0.18 (0.09–0.27)	0.17 (0.09–0.25)	0.272
	12 months after 2nd dose (V4)			
	n	54	53	
	Mean, 95% CI	0.57 (0.21–0.93)	0.35 (0.19–0.50)	0.741
% IL-2/CD4	Before Vaccination (V1)			
	n	52	52	
	Mean, 95% CI	0.033 (0.031–0.058)	0.029 (0.023–0.046)	0.034
	14 days after 2nd dose (V2a)			
	n	52	52	
	Mean, 95% CI	0.118 (0.089–0.148)	0.176 (0.134–0.218)	0.053
	6 months after 2nd dose (V3b)			
	n	54	53	
	Mean, 95% CI	0.075 (0.053–0.097)	0.095 (0.069–0.12)	0.038
	12 months after 2nd dose (V4)			
	n	54	53	
	Mean, 95% CI	0.10 (0.055–0.150)	0.11 (0.070–0.150)	0.501
% IL-4/CD4	Before Vaccination (V1)			
	n	52	52	
	Mean, 95% CI	0.204 (0.125–0.283)	0.297 (0.187–0.407)	0.380
	14 days after 2nd dose (V2a)			
	n	52	52	
	Mean, 95% CI	0.425 (0.264–0.586)	0.404 (0.220–0.588)	0.367
	6 months after 2nd dose (V3b)			
	n	54	53	
	Mean, 95% CI	0.261 (0.161–0.361)	0.295 (0.174–0.415)	0.792
	12 months after 2nd dose (V4)			
	n	54	53	
	Mean, 95% CI	0.975 (0.374–1.580)	0.699 (0.419–0.980)	0.445
% IFNγ/CD8	Before Vaccination (V1)			
	n	52	52	
	Mean, 95% CI	1.98 (1.17–2.79)	1.49 (0.70–2.29)	0.068
	14 days after 2nd dose (V2a)			
	n	52	52	
	Mean, 95% CI	2.50 (1.83–3.16)	1.82 (1.22–2.42)	0.102
	6 months after 2nd dose (V3b)			
	n	54	53	
	Mean, 95% CI	0.29 (0.11–0.48)	0.25 (0.09–0.41)	0.820
	12 months after 2nd dose (V4)			
	n	54	53	
	Mean, 95% CI	0.69 (0.01–1.37)	0.30 (0.19–0.42)	0.470
% IL-2/CD8	Before Vaccination (V1)			
	n	52	52	
	Mean, 95% CI	0.011 (0.005–0.017)	0.014 (0.006–0.207)	0.301
	14 days after 2nd dose (V2a)			
	n	52	52	
	Mean, 95% CI	0.327 (0.244–0.898)	0.399 (0.329–1.127)	0.536
	6 months after 2nd dose (V3b)			
	n	54	53	
	Mean, 95% CI	0.054 (0.018–0.089)	0.097 (0.051–0.142)	0.536
	12 months after 2nd dose (V4)			
	n	54	53	
	Mean, 95% CI	0.088 (0.054–0.123)	0.105 (0.060–0.150)	0.394

Comparative analysis was conducted using Mann–Whitney test, with a significance level set at *p*-value < 0.05.

## Data Availability

All data generated or analyzed during this study are included in this published article.

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
