# Peer review of "Immunogenicity Assessment of the SARS-CoV-2 Protein Subunit Recombinant Vaccine (CoV2-IB 0322) in a Substudy of a Phase 3 Trial in Indonesia"

_vaccines, 2024, doi:10.3390/vaccines12040371_

Round 1
Reviewer 1 Report
Comments and Suggestions for Authors
This study implemented an observer-blind, positive-control methodology to evaluate the immunogenic responses, encompassing both humoral and cellular immunity, elicited by a novel recombinant protein COVID-19 vaccine derived from a prototype strain. While the paper furnishes substantial data on the immunological responses induced by the vaccine, its significance is constrained due to the lack of distinctive attributes of the vaccine under study. Overall, the manuscript is well-compiled, albeit with room for enhancement in certain areas.
Major Comments
1. Title Section. 1) Given the limited scale of the study with only 120 subjects and its focus on immunogenicity assessment, emphasizing phase 3 in the title does not seem warranted even though this work is a segment of a phase 3 clinical endeavor; 2) The use of the term "efficacy" is inadvisable, as the manuscript exclusively addresses immunogenicity; 3) The recurrent mention of CoV2-IB 0322 requires justification; 4) Including Depok, Indonesia, in the title appears superfluous. In conclusion, a revision of the title drawing upon analogous research titles is advisable.
2. Abstract Section. 1) Given the absence of safety outcomes in the results, the mention of safety assessments in both the abstract and methods sections may need reconsideration.
3. Introduction Section 1) The contents at the outset of the first paragraph (lines 41-43) redundantly echo the initial paragraph of the discussion (lines 237 to 239); 2) A reiteration of vaccine introduction in the third paragraph (lines 62 to 70) and section 2.3 (lines 139 to 144) is observed; 3) An elaboration on the current state of the pandemic, viral variants, existing COVID-19 vaccines, and the merits of recombinant protein vaccines is recommended.
4. Materials and Methods Section 1) Given the revealed gender imbalance across groups in the results, an explanation regarding the randomization process utilized is warranted. 2) The maintenance of observer-blindness needs explicit description.
5. Results Section 1) With immunogenicity being the principal focus, the impact of prior COVID-19 infections is not negligible. As the subjects were not naive (only mild-to-moderate COVID infection within 1 month and severe infection within 3 months were excluded), further delineation of subjects' post COVID-19 infection histories, including times and timelines, in the baseline characteristics is suggested to assess balance across groups. 2) Aspects such as educational level and employment are of lesser relevance and could be condensed, in contrast, particulars like age, prevalent comorbidities, and baseline immunogenicity conditions deserve expanded details. 3) In this study, the immunogenicity seems only assessed in the ITT subset?. What is the immunogenicity subset (like ITT and PPS) defined in the protocol and SAP? Although baseline seronegative in this study does not mean naive, it might be better to assess within this population. Clarity regarding the immunogenicity subset assessment and its alignment with protocol and SAP definitions is necessitated. 4) The formats of Table 2, Table 3, and Table 4 are not standard, making the data presentation unclear. It is suggested to adjust. Authors are encouraged to consult similar studies for formatting guidance, such as this article: https://journals.asm.org/doi/10.1128/cvi.05208-11 5) The redundancy of data displayed in figures with that in tables is deemed inapt.
6. Discussion Section 1) The observed divergence in trends between neutralizing antibody levels and IgG antibody levels among the groups (notably, lower neutralizing antibodies but higher IgG for The CoV2-IB 0322 vaccine) prompts a need for exploration of underlying reasons and implications. 2) Further discussion on this study's implications and prospective avenues for COVID-19 vaccine research is recommended.
Minor Comments
1. Clarification is sought whether GMT (Dilution) or GMT (IU/mL) was employed in Figure 2, and likewise, whether IgG (AU/mL) or IgG (BAU/mL) was used in Figure 3.
2. The significance of red and blue colors in Figure 3 needs definition.
3. For Figures 4 and 5, replacing V1/V2a/V3b/V4 with corresponding timeframes (baseline/14 days/ 6 months/ 12 months) is advisable.
4. A consistent stylistic approach across all figures is recommended for enhanced visual coherence. Including a delineation for the positive cutoff value in the figures is also suggested.
Comments on the Quality of English LanguageIt is suggested to polish your English expressions a little more
Author Response
Thank you for inviting us to submit a revised draft of our manuscript. We also appreciate the time and effort you have dedicated to providing insightful feedback on ways to strengthen our paper.
Thus, it is with great pleasure that we resubmit our article for further consideration. We have incorporated changes that reflect the detailed suggestions you have graciously provided. We also hope that our edits and the responses we provide below satisfactorily address all the issues and concerns you and the reviewers have noted. To facilitate your review of our revisions, the following PDF is a point-by-point response to the questions and comments delivered in your letter.

Reviewer 2 Report
Comments and Suggestions for Authors
1. Abbreviate BMT at the start of the manuscript
2. In line 107, the manuscript misprinted the number 20 instead of 60 subjects per arm receive the SARSCoV-2 subunit.
3. The method sections for measuring the IgG and neutralizing antibodies were missing. The lack of the methods section is a significant flaw in this manuscript.
4. The methods sections also failed to mention the flow cytometry procedures. Therefore, it is impossible to interpret the CD4 and CD8 results.
5. All the tables in the manuscript were hard to read. Reformat the tables with borders for each entry to make it easy to follow each section.
6. Abbreviate IU/ml
7. Abbreviate BAU/ml
8. The table numbers were incorrectly cited in lines 189, 203, and 228. Fix them to cite correct tables and figures.
9. In Figure 2, what do the data points in the bar graph represent?
10. In Figure 3, what does the blue and red bar graph represent?
11. In section 3.3, the flow cytometry description is lacking in the intercellular cytokines results section. What does % IFN gamma/CD4 represent? Is the percentage calculated based on the previously gated population or total CD45 position populations? Elaborate the descriptions along with the missing flow cytometry methods section.
12. In line 224, the statement “An increase of IL-2 and IL-4-secreting-CD4+ T cells” contrasts the results in Table 4. The % IL-2/CD4 before vaccination values were 3.329 (CoV2-IB 0322)/ 2.858 (COVOVAX), and on day 14, the % IL-1/CD4 decreased to 0.1183 (CoV2-IB 0322)/ 0.1761 (COVOVAX). Similarly, the values in Table 4 do not correspond to those shown in Figure 4. Explain?
13. The statement in line 297, “Prior to vaccination, the COVOVAX™ Vaccine group showed higher production of 297 IFN-γ and IL-2 by CD4 cells compared to The CoV2-IB 0322 vaccine,” contrasts the results shown in Table 4 and Figure 4. Explain?
Comments on the Quality of English LanguageA few spelling mistakes and run of sentences were found.
Author Response

(The authors gave the same response as above.)

Round 2
Reviewer 1 Report
Comments and Suggestions for Authors
The author has made improvements to the manuscript based on our review comments, which have significantly enhanced the overall quality of the work. I believe it now essentially meets journal’s requirements.
Major comments:
1. Has the research been registered on websites such as clinicaltrials.gov? Please provide the trial registration number in the article.
Minor comments:
1. In the column labeled "time point" in Table 2 and Table 3, the entry "Before Vaccination (V1)" should be retained.
2. In Table 4, within the "parameter" column, I only see "% IL-4/CD4" listed. Have the other parameters been removed?
Author Response
Major comments:
1. Has the research been registered on websites such as clinicaltrials.gov? Please provide the trial registration number in the article.
Reply: I have updated the article to include the trial registration number (NCT05433285). It can now be found in lines 96-98. Thank you for pointing out this oversight.
Minor comments:
1. In the column labeled "time point" in Table 2 and Table 3, the entry "Before Vaccination (V1)" should be retained.
Reply: I'm a bit confused by the request to retain the entry 'Before Vaccination (V1)' in the 'time point' column of Table 2 and Table 3, as this label has always been included and was never removed. To clarify and ensure there are no misunderstandings, I have attached pictures of both tables showing the 'Before Vaccination (V1)' entry present in the specified location. Could you please double-check and let me know if there are any other concerns or if I might have misunderstood the assignment?

2. In Table 4, within the "parameter" column, I only see "% IL-4/CD4" listed. Have the other parameters been removed?
Reply: No parameters have been removed from Table 4. In the 'parameter' column, there are indeed 5 parameters listed, which include: % IFNγ/CD4, % IL-2/CD4, % IL-4/CD4, % IFNγ/CD8, % IL-2/CD8. I've attached the figure of Table 4 for your reference, showing all the parameters as mentioned. Please review it and let me know if there are any further concerns or clarifications needed.